# Dynamic Magnetoelectric Effect of Soft Layered Composites with a Magnetic Elastomer

**DOI:** 10.3390/polym15102262

**Published:** 2023-05-10

**Authors:** Liudmila A. Makarova, Iuliia A. Alekhina, Marat F. Khairullin, Rodion A. Makarin, Nikolai S. Perov

**Affiliations:** 1Faculty of Physics, Lomonosov Moscow State University, 119991 Moscow, Russia; ya.alekhina@physics.msu.ru (I.A.A.); khajrullin@gmail.com (M.F.K.); rodion.makaron@gmail.com (R.A.M.); perov@magn.ru (N.S.P.); 2REC SMBA, Immanuel Kant Baltic Federal University, 236041 Kaliningrad, Russia

**Keywords:** magnetoactive elastomer, magnetoelectric effect, multiferroic, iron particles, piezopolymer, layered structure, bending deformation

## Abstract

Multilayered magnetoelectric materials are of great interest for investigations due to their unique tuneable properties and giant values of magnetoelectric effect. The flexible layered structures consisting of soft components can reveal lower values of the resonant frequency for the dynamic magnetoelectric effect appearing in bending deformation mode. The double-layered structure based on the piezoelectric polymer polyvinylidene fluoride and a magnetoactive elastomer (MAE) with carbonyl iron particles in a cantilever configuration was investigated in this work. The gradient AC magnetic field was applied to the structure, causing the bending of the sample due to the attraction acting on the magnetic component. The resonant enhancement of the magnetoelectric effect was observed. The main resonant frequency for the samples depended on the MAE properties, namely, their thickness and concentration of iron particles, and was 156–163 Hz for a 0.3 mm MAE layer and 50–72 Hz for a 3 mm MAE layer; the resonant frequency depended on bias DC magnetic field as well. The results obtained can extend the application area of these devices for energy harvesting.

## 1. Introduction

Magnetoelectric materials refer to multiferroic materials, namely, materials that exhibit two or more types of ferro-ordering (ferromagnetic/antiferromagnetic, ferroelectric, ferroelastic) and transform one type of energy to another [1,2,3,4,5,6,7,8]. In such materials the latter is usually observed as a so-called magnetoelectric effect (MEE)—the appearance of electrical polarization in the material under an external magnetic field—as well as the inverse effect [9]. Composite multiferroics of different geometries, such as multilayers, nanocomposites, carrier-mediated composites with ferromagnetic and ferroelectric inclusions, etc., were reported to show the magnetoelectric transformation with the largest magnitudes [10,11,12,13,14], which exceeded those for single-phase multiferroics by several orders of magnitude [15,16]. Moreover, their properties can be tuned in order to meet the requirements of specific engineering or biomedical applications [17,18,19,20].

Among composite structures, multiferroic multilayers demonstrate one of the largest values of the magnetoelectric coefficient [21,22]. In the examples of this class of materials, the mechanism of the magnetoelectric coupling is associated with an interconnection between the magnetostriction and piezoelectric effect by means of mechanical connections. The magnitude of the MEE can be sufficiently increased in the AC regime as a result of the mechanical resonances of the whole structure: in [23], it was demonstrated that the mechanical oscillations of a magnetostrictive layer under an AC longitudinal magnetic field caused the appearance of a resonant electrical polarization due to the piezoelectric effect.

Despite the intense investigations in the field of multilayered multiferroics, numerous questions such as properties modifications still require more detailed investigations and thus remain the focus of research [24]. In particular, in [20,25,26,27,28,29], the authors mentioned that the utilization of polymer components in a magnetoelectric structure should be promising for the development of flexible electronics including sensors and energy-harvesting devices or biomedical devices. The softness of the structure based on polymers allows a reduction of the resonant frequency, which is of great importance for biomedical and energy-harvesting applications [27]. For this reason, the investigations of layered structures based on magnetic elastomers and piezoelectric polymers (PEP) are of particular interest.

Magnetic elastomers (magnetoactive elastomers, MAE), which are suggested for utilization as one of the components of soft multiferroics, consist of a polymer matrix with filling ferromagnetic (FM) micro- or nanoparticles [29,30]. The peculiar properties of magnetic elastomers in magnetic field appear due to the dipolar interactions of magnetized filling particles [31]; in particular, a rearrangement of ferromagnetic filler in an elastic medium is responsible for the observations of magnetodeformation, magnetorheological effect and so on [32,33]. The particles’ displacement also accounts for the observation of a hysteresis loop widening and susceptibility maxima in nonzero magnetic fields for soft magnetic elastomers (with a Young’s modulus of the order of 10^4^ Pa) [34]. In the vicinity of these field values, a maximum sensitivity of magnetodeformation to the applied magnetic field is likely to be observed.

The origin of the magnetoelectric transformation in the structure MAE-PEP is the bending of the magnetoelastomeric layer in a gradient magnetic field [25]. The bending occurs due to the force acting on the magnetic elastomer from a gradient magnetic field. This deformation is transmitted to the piezoelectric layer with the appearance of a voltage. The above-described displacement of the particles and interparticle interactions in the soft matrix can also contribute to this mechanism because of the susceptibility modulation as well as its influence on the multilayer mechanical properties.

The configuration of electric and magnetic fields can be varied in relation to the plane of layers. The authors of the work [26] studied the MEE in the same layered structure in a uniform-pulse magnetic field. In this geometry, the initial bending of the sample in a zero magnetic field under the effect of gravity and a short pulsed field led to a gradient field distribution inside the sample after switching the magnetic field on, which resulted in a further deformation of the sample. In another work [27], the AC magnetic field was applied perpendicular to the sample plane, and a DC uniform magnetic field was applied in the longitudinal direction to the sample plane. The AC magnetic field was used to cause the resonant oscillations of the structure.

Investigations of the MEE of the layered-structure MAE-PEP for a cantilever configuration were carried out. Earlier, the series of investigations of the MEE under magnetic field pulses of different shapes were conducted [25]. Firstly, the external magnetic field was switched on and off and the sample had damping oscillations. In the second step, the dependence of the induced voltage on the duration of the pulse was investigated. Here, we present the continuation of this research, where the AC magnetic field was applied to the sample.

The goal of the work was to investigate the dynamic magnetoelectric effect in a layered-structure magnetic elastomer and piezopolymer, namely, the detection of the resonant amplification of the induced signal at an applied gradient AC magnetic field. The frequency sweep of the AC field allowed us to investigate the resonant behaviour of the multilayer. The setup was upgraded for the simultaneous and synchronized supply and measurements of AC signals. In addition, the search of the dependence of the resonant frequency on the type of MAE layer was carried out. MAEs with different mechanical and magnetic characteristics were used for the multilayers’ creation. The investigations were aimed at the search for the influence of a DC uniform magnetic bias field on the resonant frequency and induced signal as well. The application of the gradient magnetic field made it possible to observe pronounced bending deformations of the samples, so it remained unchanged. The transverse configuration of magnetic and electric signals (opposite to the plane of the multilayer) was utilized.

The results of the research could be promising for the development of sensors of different types, for example, a field inhomogeneity sensor, a wide-range vibration sensor, a vibration-energy-harvesting devices, elements for utilization in biomedical applications.

## 2. Materials and Methods

The samples of MAE-PEP and the method of their preparation were described in the a previous work [25]. The PEP was the commercially available piezoelectric polyvinylidene fluoride (PVDF) with conductive plates. The layers of MAE contained the carbonyl iron microparticles (IP) with an isotropic distribution inside the polymer matrix. The well-established manufacturing protocol was utilized for the production of the samples, which guaranteed the quality of the samples and a uniform filler distribution in the MAE. The carbonyl iron particles were used as filling particles as they have high values of magnetic susceptibility and saturation magnetization for the materials, and it can result in an increase of the sample response to external magnetic field. The characterization of the particles’ size distribution was carried out with a transmission electron microscope (TEM); the iron particles had an almost spherical shape, and their size distribution was from 0.5 to 5 μm [25]. The dynamic measurements of the MEE were carried out with the “thick” MAE samples (the thickness of the MAE layer was 3 mm, the concentrations of iron particles were 40, 65, 73, 77 wt%) and “thin” MAE samples (the thickness of the MAE layer was 0.3 mm, the concentrations of iron particles were 56, 65, 70, 75, 80 wt%). The Young’s moduli of the “thick” and “thin” elastomers were different (Table 1 in [25], Table 1). The thickness of the piezoelectric polymer between the conductive plates was 28 µm. For the details of the preparation process and the explanation of the material choice, refer to [25].

The experimental setup for the measurement of the voltage induced in the PVDF layer due to the deformation of the composite sample under an external magnetic field was modified. The cantilever configuration, namely, the configuration of the sample with one fixed end, was used (Figure 1a). The length of the free end of the sample was L = 15 mm, the width was 13 mm. The AC magnetic field was created by one electromagnet, so the field was spatially gradient. The application of a gradient magnetic field resulted in a pronounced bending deformation of the sample. The distance between the undeformed sample plane and electromagnet surface was 3 mm, which allowed us to create a magnetic field of a magnitude which was enough for a pronounced deformation of the sample and at the same time a bending of the sample occurred without any mechanical contact with the electromagnet surface. The measurements of the time dependence of the electrical current in the electromagnet and the induced electrical voltage in the PEP were carried out by two different microcontrollers. The synchronization and frequency control were conducted by the third microcontroller. The schematic presentation of the setup is shown in Figure 2. The frequency range was up to 200 Hz with 2 Hz increments. The amplitude of the AC magnetic field in the centre of the free edge of the sample varied from 0 to 132 Oe with 22 Oe increments.

The test of the influence of the bias field on the dynamic MEE in the MAE-PEP composites was also carried out. The direction of the bias DC field created by the permanent magnet was perpendicular to the sample plane, as it is presented in Figure 1b. Thus, the AC and DC magnetic fields were collinear. The amplitude of the AC magnetic field was 110 Oe, the value of the DC homogeneous magnetic field was up to 800 Oe.

## 3. Results

The frequency dependence of the voltage induced under the influence of an AC magnetic field with different amplitudes was measured for all samples. The typical dependence represented by those amplitudes for the sample with a thin MAE layer with a 70 wt% IP concentration is shown in Figure 3. The curves had the main resonance peak at the frequencies near 163–172 Hz and several resonance-like peaks with a lower amplitude at submultiple frequencies. For example, at a fixed field amplitude, the main resonant frequency for the mentioned sample was ω_1_ = 162 Hz, the other resonant frequencies were ω_1_/2 = 83 Hz, ω_1_/3 = 55.4 Hz, ω_1_/4 = 41.4 Hz and ω_1_/5 = 33.4 Hz.

The mechanical resonance of the bending deformations allowed us to significantly enhance the induced electrical signal compared with the values obtained in the quasistatic mode—the increase reached one hundred percent. For example, the value of the induced voltage for the presented sample (Figure 3) was larger by factors of 9.6 and 3.7 compared with the first (sample bending) and second (sample unbending) voltage peaks obtained in the quasistatic regime, correspondingly (52.9 mV in the AC field with an amplitude of 132 Oe at the resonance frequency vs. 5.5 mV when switching the magnetic field on and 14.1 mV when switching it off) [25]. The resonant frequency slightly changed with the changes of amplitudes of the AC magnetic field, which was probably caused by different field gradient values for its different amplitudes. Namely, the resonant frequency shifted from 172 Hz to 163 Hz at an increase of the magnetic field amplitude from 22 Oe to 132 Oe.

The analysis of the influence of the MAE layer properties on the magnetoelectric effect can be performed by the comparison of the frequency dependence of the voltage induced by the magnetic field of a fixed amplitude 110 Oe for the samples with different IP concentrations, which is presented in Figure 4 for thin (Figure 4a) and thick (Figure 4b) MAE layers.

The value of the induced signal depended on the IP concentration in the MAE layer for the thin and thick cases. The magnetic moment of the MAE sample increased with the increase of the IP concentration, which resulted in an increment of the acting force that was proportional to the magnetic moment, and in a corresponding increment of the deformation amplitude.

The values of the main resonant frequencies for different samples are presented in Table 1. The resonant frequencies had a tendency to decrease with the increase of the IP concentration in the MAE layer. Considering the movement described by the model of a beam bending, this could be associated with the dependence of the natural oscillation frequency on the effective Young’s modulus and effective density of the structure [25,26] assuming other geometrical parameters to be the same. Both material’s parameters increased with the IP concentration in the MAE layer, although, the dependence on the filler content was nonmonotonic for thin MAE layers, which could be associated with nonlinearities appearing in oscillations with large amplitudes.

The resonant frequency was lower for the samples with a thick MAE layer if the iron concentration remained unchanged, which was in agreement with the model of a beam bending, as the frequency is inversely proportional to the thickness of the beam. For example, for two samples with the same IP concentration 65 wt% and different thicknesses, the resonant frequencies were 158 Hz for the thin MAE layer and 72 Hz for the thick MAE layer. However, the value of the induced signal was larger for the sample with the thick MAE layer and reached 115 mV, while for the thin MAE layer, it was about 21.4 mV at the same magnetic field amplitude.

The mean value of the magnetoelectric effect was calculated as the induced voltage divided by the thickness of the piezopolymer layer between conductive plates (28 µm) and the value of the magnetic field amplitude at the centre of the free edge of the sample. The maximum value of the MEE (mean value) reached 0.77 V/cm*Oe.

The modulation of the dynamic MEE in the composite by the applied bias field was also investigated for the considered samples. The representative results for the composite with a thin MAE with 80 wt% IP are presented in Figure 5. The magnetic moment of the MAE sample increased with the additional application of a DC magnetic field, which resulted in an increase of the acting bending force and thus, in a signal enhancement. For instance, the value of the induced signal for the presented sample increased by 620% in the applied field of 800 Oe. The mean value of the MEE increased from 0.11 V/cm*Oe for the zero bias field to 0.76 V/cm*Oe for the 800 Oe bias field. A rise of all resonant-like peaks was also observed. A shift of resonant frequencies with the bias field application was observed as well: for the results presented in Figure 5, the main resonant frequency shifted by 15 Hz to lower values in the 800 Oe bias field.

The influence of the bias field on the induced signal and resonant frequency was investigated. Figure 6 presents the obtained dependence, where the dots mean the experimental data and the dashed lines present its approximations.

The induced voltage linearly increased with the increase of the bias field. The linear coefficient of dependence obtained increased with the increase of the IP concentration in the MAE layer. It was in accordance with the fact that the value of the magnetic susceptibility of the MAE increased with the increase of the IP concentration. An explanation of the linear dependence of the voltage on the bias field is proposed in the Discussion section.

The dependence of the resonant frequency on the bias field was fitted by a linearly decreasing function. The data for the samples with a thin MAE layer are presented in Appendix A. The modulus of the linear coefficient increased with the increase of the IP concentration in the MAE layer. The first assumption demonstrated that the resonant frequency for such a geometrical structure depended on the effective Young’s modulus and that value increased with the increase of the IP concentration in the MAE layer at a fixed elastic modulus of the initial polymer (see Table 1).

## 4. Discussion

For all the presented above results on the frequency dependence of the dynamic MEE for two-layered structures with different types of MAE layer, we considered the oscillations to deviate from the harmonic law due to the inhomogeneity of the external magnetic field. The force acting on the sample was not constant over the oscillations period, as it depended on both the distance between the sample and the electromagnet, and on the magnetic field strength, which varied with the sample bending in the geometry of the experiment. In addition, the oscillations could not be described as small, as the sample deflection was comparable with its dimensions. However, the model of beam bending described the main patterns in the dependence of the effect on the sample properties and external influences. The estimation of the value of the resonant frequency according to the model [27,35] allowed us to infer that there was only the first bending mode in the presented structure at the main frequency, as shown in Figure 1. We considered the only bending oscillation mode in the investigated frequency range, and we neglected the longitudinal oscillation modes due to the configurations of the measurements as well as to the structure of the samples with a possible slipping of the layers along the interface.

The obtained resonant frequencies were close to the values of the natural oscillation frequencies obtained in the quasistatic regime [25]; the comparison was presented in Table 1. The slight quantitative differences were associated with the influence of the gradient magnetic field inhomogeneity and nonharmonic oscillations of the sample.

Moreover, there were several resonant-like peaks with lower amplitudes at submultiple frequencies. Namely, the resonant signals were observed at the frequencies ω_1_, ω_1/2_, ω_1/3_, etc., as presented in Figure 3 and Figure 4. There were five distinguishable resonances for the samples with a thin MAE layer, while there were three resonances for the samples with a thick MAE layer. It can be explained by the differences between the elastic modulus for the samples: the Young’s modulus was larger for thin MAE samples by approximately two orders of magnitude then for thick MAE samples, which influenced the loss factors during mechanical deformations. The appearance of resonant-like peaks at multiples of the frequencies meant the acting force was applied two, three, etc., times less often than for the main resonance. A rarer application of the acting force led to a smaller induced signal. However, all resonances observed in that frequency range were associated with the first bending mode of the sample, as shown in Figure 1.

Under the applied external DC magnetic field, the bias field, the MAE’s Young’s modulus increased due to magnetorheological effect, and the density of the materials and the mass distribution changed due to the magnetodeformation. The magnitude of the effect increased as the magnetization of the MAE in the sample composition increased, leading to the enhancement of the acting bending force and, as a consequence, to an increment on the amplitude and the rate of deformation. As it was previously discussed [25], the strain rate also affected the magnitude of the signal. It was associated with the fact that the stress in the piezoelectric layer occurred only during deformation but not in a static bent state.

We expected the induced signal to depend on the bias magnetic field via a linear function as suggested in the following:(1)dU~dx
where dU is the induced voltage at the piezoelectric element and dx is the elementary deformation of the sample. The bending deformation can be presented at two elementary deformations of two nominal layers—the stress and strain deformations. According to the elastic law,
(2)dx~F,
where F is the acting force on the sample. The force acting on the piezolayer is equal to the force acting on the MAE layer from a gradient magnetic field h and
(3)$$F\sim M\;gradh\sim \mu \left(H\right)H\;*\;gradh,$$
where M is the magnetic moment of the MAE sample in the effective magnetic field H and $\mu \left(H\right)$ is the permeability, which depends on the magnetic field. The value gradh is the constant for a fixed amplitude of the AC magnetic field. Thus, the induced signal is
(4)$$dU\sim \mu \left(H\right)H\;*\;gradh$$

The integrated induced voltage finally depended on the permeability of the sample and bias magnetic field. With an increased magnetic field, the permeability of the MAE decreased, although the final dependence was increasing. This means that the presented experimental data were in such a field region, where the decreasing permeability affected the signal less than the increasing bias field.

The results were compared with the data presented in the work [27], where an AC homogeneous magnetic field with an amplitude of 2.7 Oe caused the oscillations of the PEP-MAE sample (the thickness of the MAE varied from 0.85 mm to 3.01 mm and the IP concentration was 80 wt%). The value of the resonant MEE without any bias field for the sample with an MAE thickness of 2.77 mm was measured at the frequency 22.7 Hz and was 2.67 V/cm*Oe. In our studies, the resonant frequencies were 60 Hz, 33 Hz and 21 Hz, and the mean value of the MEE at the main frequency was 0.77 V/cm*Oe for the sample with an MAE thickness of 3 mm and an IP concentration of 77 wt%. The results were qualitatively consistent, and the quantitative difference was associated with the different geometries of the experiment. A uniform magnetic field was utilized by the authors of [26,27], whilst current research is focused on inhomogeneous AC magnetic field applications for sample bending.

In an external uniform magnetic field, several parameters such as the effective Young’s modulus, effective density and mass distribution changed, and the mechanical deformations of the MAE occurred. The joint influence of the changes in mechanical properties resulted in a bias of the resonance frequency. An analytical model was not suggested, as it required some changes in the initial equation of motion of a bending beam.

The results above can extend the application area of devices for energy harvesting. For example, the additional peaks in the spectra can be used for sensors for vibration control at different frequencies, namely, a wide-range vibration sensor. The obtained dependence of the resonant frequency on the amplitude of the AC magnetic field can be used for developing structures such as sensors of field inhomogeneity. Such types of oscillations of the PEP-MAE structure allow us to develop different elements for real external conditions with scattered inhomogeneous magnetic fields. Further research on the dependence of the effect on the magnitude of the field gradient will demonstrate the potential of this method of detection.

## 5. Conclusions

The magnetoelectric effect in a double-layered structure based on a piezoelectric polymer and a magnetoactive elastomer with iron microparticles was investigated. A distinctive peculiarity of the research was the use of an inhomogeneous AC magnetic field to induce a nonlinear bending mode of the sample. The frequency dependence of the voltage induced in the PEP-MAE structure had a main resonance peak at the frequencies near 156–163 Hz for the thin (0.3 mm) MAE layer and 50–72 Hz for the thick (3 mm) MAE and several resonance-like peaks with lower amplitude at submultiple frequencies. The values of the signal and the resonant frequency strongly depended on the bias uniform magnetic field, and the dependence had a linear behaviour. The variations of the resonant frequency were obtained with changes in the amplitude of the AC magnetic field as well. The value of the MEE in the presented results was lower than reported in the literature, but the differences between the geometry of the experiments and our focus on the inhomogeneous AC magnetic field application can explain it. The maximum value of the mean MEE was observed for a sample with a 3 mm thickness of the MAE layer and a 77 wt% IP concentration and was 0.77 V/cm*Oe.

## Figures and Tables

**Figure 1 polymers-15-02262-f001:**
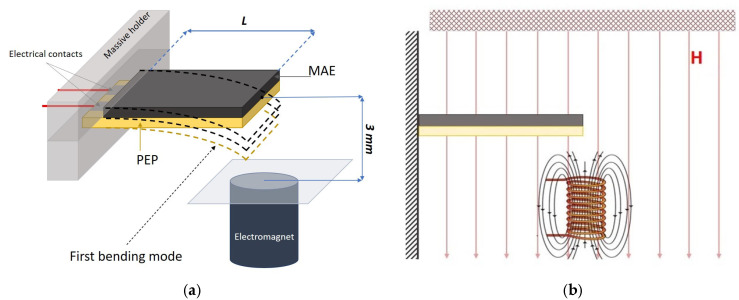
(**a**) The cantilever configuration of the two-layered sample; the distance between the electromagnet and sample plane is fixed at 3 mm. The length of the free end of the sample is L. (**b**) The measurement configuration of the sample in the gradient AC magnetic field generated by the electromagnet and a uniform DC magnetic field, which is perpendicular to the sample plane.

**Figure 2 polymers-15-02262-f002:**
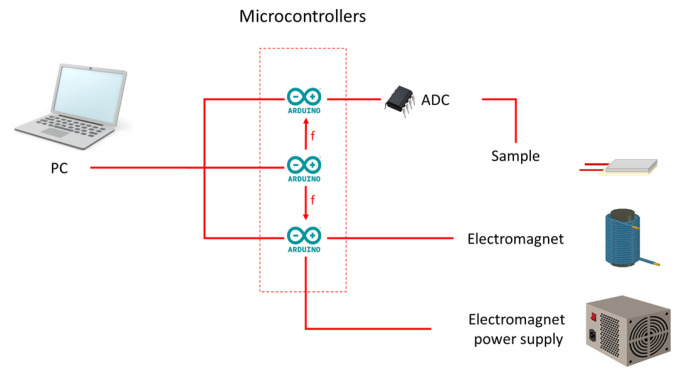
The schematic presentation of the measurement setup with three microcontrollers.

**Figure 3 polymers-15-02262-f003:**
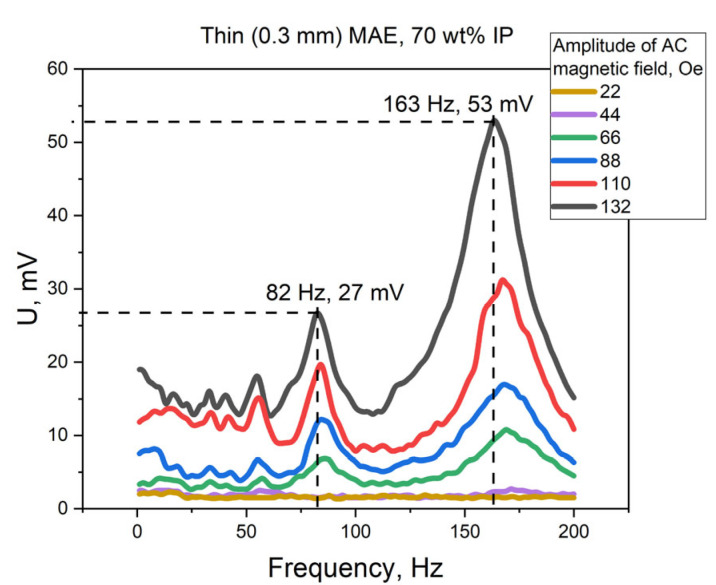
The frequency dependence of the voltage induced under the influence of an AC magnetic field with different amplitudes for the sample with a thin MAE layer and a 70 wt% IP concentration.

**Figure 4 polymers-15-02262-f004:**
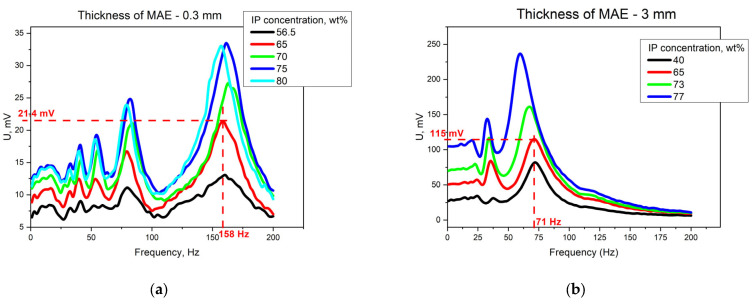
The frequency dependence of the voltage induced under the influence of an AC magnetic field with an amplitude of 132 Oe for different samples with (**a**) a thin MAE layer and (**b**) a thick MAE layer.

**Figure 5 polymers-15-02262-f005:**
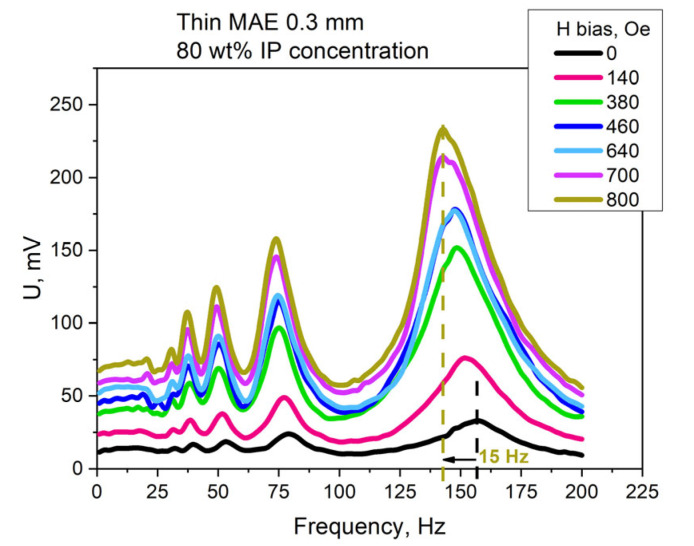
The frequency dependence of the MEE under the influence of the AC magnetic field with an amplitude of 110 Oe and with an applied uniform bias magnetic field of up to 800 Oe for the sample with a thin MAE layer and an 80 wt% IP concentration.

**Figure 6 polymers-15-02262-f006:**
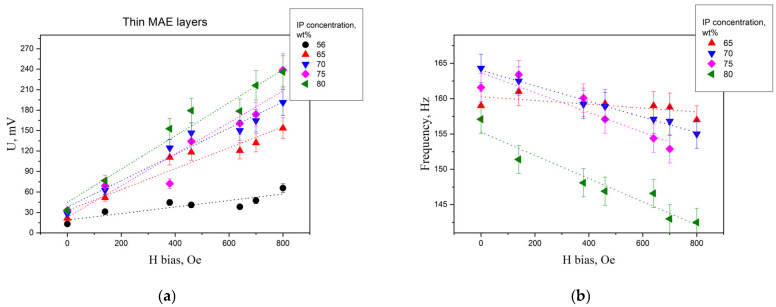
The dependence of (**a**) the voltage induced and (**b**) the main resonant frequency on the uniform DC magnetic field bias for the sample with a thin 0.3 mm MAE layer with different IP concentrations. The values of the fitting lines’ slope with errors and their R-square coefficients are presented in Appendix A.

**Table 1 polymers-15-02262-t001:** Data on the resonance frequency, natural frequency and mean value of the magnetoelectric effect for the samples with different thicknesses of the MAE layer and IP concentrations at the AC magnetic field amplitude 110 Oe. Reprinted with permission from Ref. [25]. 2020, IOP Science.

IP Concentration in MAE Layer	Young’s Modulus, kPa [21]	Main Resonance Frequency, Hz	Natural Frequency, Hz [21]	Mean Value of MEE, mV/cm*Oe
	The thickness of the MAE is 0.3 mm			
56	1500	160	117	42.5
65	1600	158	142	69.5
70	2200	163	123	88.6
75	2500	161	123	109.1
80	6700	156	87	107.5
	The thickness of the MAE is 3 mm			
40	51	72	69	266.9
65	59	71	66	373.4
73	65	67	60	522.1
77	90	60	57	770.4

## Data Availability

All the necessary data is given in the article and Appendix A.

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
