# Peer review of "Dynamic Magnetoelectric Effect of Soft Layered Composites with a Magnetic Elastomer"

_polymers, 2023, doi:10.3390/polym15102262_

Round 1

Reviewer 1 Report

Submitted version of the manuscript is very badly prepared. It containes non-defined abbreviations, lost intervals, los commas even los figure 1. Introduction is close to be reasonable but it contains not typical expressions like "ferro-ordering" (ferromagnetic/antiferromagnetic?). There is no goal of the stydy at the end of the Introduction. One can find long step by step desctiption of selected parts of the studies and even reference to previous work. At the end with a great surprise audience can read part belonging to conclusions without any support from the experimental data. 

Materials and method section make reference to previous work of the same group and even Table is expected to be analysed from https://iopscience.iop.org/article/10.1088/1361-6463/abb7b7/pdf contribution. Authors must give most important details here and explain why they use selected materials and measurements conditions. For example, what happens if the carbonil iron will be substituted by iron nanoparticles of about 50 nm in diameter? There is no description/photograph of the cantilever configuration or the reason to use particular size of the samples (depending on their length?).

References are limited to a very few groups and do not represent the contributions of the major experts. In addition, the type of the resonance is not clearly defined, it is confusing for young readers. MDPI journals contributions are not properly represented. 

Acceptable

Author Response

We sincerely thank the reviewers for the corrections and comments made. We believe that the revision will improve the presentation of the results. The list of the changes made according to suggestions of the Reviewers is summarized below.

Reviewer #1

Submitted version of the manuscript is very badly prepared. 

We apologize for the first impression from our work and we tried to improve the work according to your comments.

1) It containes non-defined abbreviations, lost intervals, los commas even los figure 1.

We thank the reviewer for the comment. We double-checked all abbreviations and figures in the text. We improved the punctuation as well, however, some of the lost intervals might have appeared due to the conflicting software. We tried to correct all the found mistakes and typos. 

2) Introduction is close to be reasonable but it contains not typical expressions like "ferro-ordering" (ferromagnetic/antiferromagnetic?). There is no goal of the stydy at the end of the Introduction. One can find long step by step desctiption of selected parts of the studies and even reference to previous work. At the end with a great surprise audience can read part belonging to conclusions without any support from the experimental data. 

The expression “ferro-ordering” was modified by additive clarification: “ferro-ordering (ferromagnetic/antiferromagnetic, ferroelectric, ferroelastic)”. The goal of the study was formulated and explained in more detail. We correct the last paragraph in the Introduction according to assumption about the application of our results. At the end of Discussion there was the more detailed description of applications of results obtained.

3) Materials and method section make reference to previous work of the same group and even Table is expected to be analysed from https://iopscience.iop.org/article/10.1088/1361-6463/abb7b7/pdf contribution. Authors must give most important details here and explain why they use selected materials and measurements conditions. For example, what happens if the carbonil iron will be substituted by iron nanoparticles of about 50 nm in diameter? There is no description/photograph of the cantilever configuration or the reason to use particular size of the samples (depending on their length?).

We reproduced the data from previous work in the present work with the appropriate citing marks, so that the explanation of the results would be coherent and understanding of the  the work would not require additional search in literature. We added the description of cantilever configuration with corresponding schematic image. Also we clarified the utilization of carbonyl iron particles as filling particles in magnetic elastomers and explained the measurement setup and conditions.

“The carbonyl iron particles were used as a filling particles as they have high values of magnetic susceptibility and saturation magnetization for the materials, and it can result in increase of the sample response to external magnetic field.”

Magnetic elastomers with particle sizes of 5 µm have some of the best manufacturing parameters. The particles are not too large, so the uniformity and isotropy of their distribution in the matrix are observed. The use of nanoparticles introduces difficulties in the manufacture of elastomers due to their high surface area to volume ratio.

“Experimental setup for measurement of the voltage induced in PVDF layer due to deformation of the composite sample under external magnetic field was modified. The cantilever configuration, namely, the configuration of the sample with one fixed end, was used (Fig.1a). The length of the free end of the sample L=15 mm, the width - 13 mm. The AC magnetic field was created by one electromagnet, so the field is spatially gradient. The application of gradient magnetic field resulted in pronounced bending deformation of the sample. The distance between the undeformed sample plane and electromagnet surface was 3 mm, that allowed it to apply magnetic field with the value for pronounced deformation of the sample and at the same time allowed it to bend for the sample without mechanical contact with the electromagnet surface.”

A change in the geometric parameters of measurements, in particular, a change in the length of the free end of the cantilever, leads to a change in the resonant frequency, as well as a change in the induced signal, therefore, in the process of research, it is necessary to observe the same conditions for measuring different samples. In addition, if the distance between the surface of the electromagnet and the plane of the sample is changed, then the value of the magnetic field and its gradient will also change.

4) References are limited to a very few groups and do not represent the contributions of the major experts.

We extended the list of referenes, it includes the investigations by such experts as Gupta R., Ramesh R., Pyatakov A., Zvezdin A.K., Viehland D.S., Fetisov Y., Kholkin A., Shamonin M., Bastola A.K., Stepanov G.V., Raikher Y.L., Kramarenko E., Khokhlov A.R., etc. We were guided by correspondence of the world investigations to our introduction, namely, we would like to show the logical justification of our studyings. 

5) In addition, the type of the resonance is not clearly defined, it is confusing for young readers.

We added the clarification about the type of resonance:

“... we consider the oscillations to deviate from harmonic law due to inhomogeneity of the external magnetic field. … In addition, the oscillations cannot be described as small, as the sample deflection is comparable with its dimensions. However, the model of beam bending describes the main patterns in the dependences of the effect in the sample properties and external influences. The estimation of the value of resonant frequency according to the model [23, 31] allowed to suppose that there was only the first bending mode in the presented structure at the main frequency, as it is presented in the Fig. 1.”

6) MDPI journals contributions are not properly represented. 

We appreciate the comment from the Reviewer. The list of references includes the journals from the MDPI editorial (see, for instance, references 15, 18, 22-23). However, as the presented investigations occupy the position on the intersection of different research areas - it requires the analysis of magnetic properties, piezoelectricity, laminated composites, as well as great engineering accents in the experiment - the cited literature is widely dispersed among numerous scientific journals having different topics. For this reason, the collection of references has wide variety of journals, which are represented by different editors. In this work we chose the most representative works of the classes, however, we believe, that the development of this topic is going to be well presented in MDPI journals.

Reviewer 2 Report

Makarova et al. report in this article, with title “Dynamic magnetoelectric effect of the soft layered composites with magnetic elastomer“, on the magnetoelectric effect investigated in a double-layered structure based on a piezoelectric polymer and magnetoactive elastomer with carbonyl iron particles. The authors report that it is observed how the gradient AC magnetic field applied to this system generates its bending, due to the attraction of the magnetic component. The authors claim that the reported results can extend their application to devices for energy harvesting. Overall, the quality of the work is high enough and appropriate to be published in the journal POLYMERS.

Some important suggestions/revisions are:   

1)     In the abstract section, please, define what PVDF and MAE are, as done in section 2 (Materials and Methods).

2)     In the introduction section, please, define what FM is.

3)     The Figure 1 is missing. Please, revise that.

4)     Pages 7 and 8. The journal abbreviations of most of the references must be revised.   

 Moderate editing of English language

Author Response

We sincerely thank the reviewers for the corrections and comments made. We believe that the revision will improve the presentation of the results. The list of the changes made according to suggestions of the Reviewers is summarized below.

Reviewer #2

Makarova et al. report in this article, with title “Dynamic magnetoelectric effect of the soft layered composites with magnetic elastomer“, on the magnetoelectric effect investigated in a double-layered structure based on a piezoelectric polymer and magnetoactive elastomer with carbonyl iron particles. The authors report that it is observed how the gradient AC magnetic field applied to this system generates its bending, due to the attraction of the magnetic component. The authors claim that the reported results can extend their application to devices for energy harvesting. Overall, the quality of the work is high enough and appropriate to be published in the journal POLYMERS.

Some important suggestions/revisions are:   

1)     In the abstract section, please, define what PVDF and MAE are, as done in section 2 (Materials and Methods).

2)     In the introduction section, please, define what FM is.

3)     The Figure 1 is missing. Please, revise that.

4)     Pages 7 and 8. The journal abbreviations of most of the references must be revised.

The answer:

We thank the reviewer for the positive comments about our work. We tried to correct the points in the accordance with the comments. We carefully checked the use of abbreviations, we added the Fig 1 and we corrected the reference list. 

Besides, we improved the data presentation and added some extended data.

Round 2

Reviewer 1 Report

Submitted version of the manuscript is much better organized and the major part of questions were answered. However, there are still two problems to solve prior to possible publication.

I am sorry to say but Authors reply “Magnetic elastomers with particle sizes of 5 µm have some of the best manufacturing parameters. The particles are not too large, so the uniformity and isotropy of their distribution in the matrix are observed. The use of nanoparticles introduces difficulties in the manufacture of elastomers due to their high surface area to volume ratio” is clearly not acceptable. First of all, set of the particles under consideration has the particles distribution and it should be discussed even for commercial set. Second point is their distribution in the matrix. Nowadays there are methods for visualization of the particles size and the structure of different filled composites ( Dushin et al. DOI: 10.15593/perm.mech/2018.2.03; Safronov et al., Sensors 2018, 18(1), 257; Simonov-Emelyanov https://doi.org/10.4028/www.scientific.net/KEM.899.694 etc.). Work provides no structural characterization, no discussion of this problems and no estimation of the future trends with this respect. This is clear disadvantage.

Figure 6 shows linear dependences on H bias and Bias H – what was the difference? Estimation of the errors and quality of the linear fit must be given.

Author Response

Submitted version of the manuscript is much better organized and the major part of questions were answered. However, there are still two problems to solve prior to possible publication.

1) I am sorry to say but Authors reply “Magnetic elastomers with particle sizes of 5 µm have some of the best manufacturing parameters. The particles are not too large, so the uniformity and isotropy of their distribution in the matrix are observed. The use of nanoparticles introduces difficulties in the manufacture of elastomers due to their high surface area to volume ratio” is clearly not acceptable. First of all, set of the particles under consideration has the particles distribution and it should be discussed even for commercial set. Second point is their distribution in the matrix. Nowadays there are methods for visualization of the particles size and the structure of different filled composites ( Dushin et al. DOI: 10.15593/perm.mech/2018.2.03; Safronov et al., Sensors 2018, 18(1), 257; Simonov-Emelyanov https://doi.org/10.4028/www.scientific.net/KEM.899.694 etc.). Work provides no structural characterization, no discussion of this problems and no estimation of the future trends with this respect. This is clear disadvantage.

Answer:

We are grateful to the Reviewer for this comment. As this article presents the results obtained on the samples, which were previously described in detail, including those for preparation process, we did not include the same information in this article both no keep the focus of the paper on the measurements techniques as well as not to duplicate the presented information and avoid incorrect citings. However, we totally agree with the Reviewer, that some of the information is missing for the coherent description of the investigated materials and understanding of the paper. Here we included the following information about the characterization of MAE samples:
The well-established manufacturing protocol was utilized for the production of samples, which guaranteed the quality of the samples and uniform filler distribution in the MAE. The carbonyl iron particles were used as a filling particles as they have high values of magnetic susceptibility and saturation magnetization for the materials, and it can result in increase of the sample response to external magnetic field. Characterization of particles size distribution was carried out with transmission electron microscope TEM, the iron particles had almost spherical shape and their size distribution was from 0.5 to 5 μm [25].

For the details of preparation process and the explanation of the material choice, please, refer to the [25].”

In reference 25 we explained all the details about the materials choice and preparation process. Here we include the citation of the information, presented there:

“Carbonyl iron particles of nearly spherical shapes and 0.5-5 µm size distribution were used as a filler in all series of MAE samples. Characterization of particles size distribution was carried out with transmission electron microscope TEM, the iron particles had almost spherical shape and their size distribution was from 0.5 to 5 μm as it is shown in the Figure 2.

Different types of magnetic fillers in elastomers allow to tune their magnetic, electrical and even mechanical properties. One of the most commonly used magnetic fillers is carbonyl iron as it has one of the largest value of susceptibility and high saturation limit [https://doi.org/10.3390/ma11061040]. The other types of fillers in elastomers, for instance, nonconducting or hard magnetic particles, can result in significant changes of material properties and thus, functioning principles of devices based on it [https://doi.org/10.1002/adem.201400258, DOI: 10.1088/0964-1726/24/2/025016, DOI: 10.1088/0964-1726/24/3/035002]. To tune the properties of magnetoactive elastomers one can vary the shape and size of magnetic particles [https://doi.org/10.1016/j.compositesb.2015.09.013]. Using anisotropic magnetic particles allows to amplify the properties of MAE in one direction.”

The problem of particles’ distribution inside the matrix also was described in previous work. Namely, the method of preparation of magnetic elastomers was well-established and examined for numerous samples of magnetic elastomers with different filling particles, for example, iron particles, neodymium-iron-boron particles, barium ferrites, etc. The different sizes of particles were used as well. DOI: 10.1088/1361-665X/aa82e9, DOI: 10.1088/0953-8984/20/20/204121. In this work the investigations of particles distribution using tomography methods were not carried out and, thus, images could not be shown, however, we believe that the preparation method can guarantee the isotropic particles distribution in the polymer matrix.

2) Figure 6 shows linear dependences on H bias and Bias H – what was the difference? Estimation of the errors and quality of the linear fit must be given.

Answer: The caption “H bias” was corrected. The estimation errors and quality of the linear fit were made.